# Therapeutic Vaccines for HPV-Associated Cervical Malignancies: A Systematic Review

**DOI:** 10.3390/vaccines12040428

**Published:** 2024-04-17

**Authors:** Souhail Alouini, Chantal Pichon

**Affiliations:** 1Departement of Gynecological Surgery, Centre Hospitalier Universitaire d’Orléans, 14 Avenue de l’Hôpital, 45100 Orleans, France; 2Faculté de Médecine, Université d’Orléans, 45100 Orleans, France; 3Institut Universitaire de France, 1 rue Descartes, 75035 Paris, France; pichon.chantal@gmail.com; 4INSERM ART ARNm, University of Orléans, 45100 Orleans, France

**Keywords:** human papillomavirus, uterine cervical neoplasms, therapeutic vaccines, cervical cancers, cervical intraepithelial neoplasia

## Abstract

Importance: Despite widespread prophylactic vaccination, cervical cancer continues to be a major health problem with considerable mortality. Currently, therapeutic vaccines for HPV-associated cervical malignancies are being evaluated as a potential complement to the standard treatment. Objective: The present systematic review was conducted on randomized controlled trials (RCTs) to investigate the effects of therapeutic vaccines on the treatment of patients with cervical cancer and cervical intraepithelial neoplasia (CIN) of Grades 2 and 3. Evidence Review: The PubMed, Embase, and Cochrane Central Register of Controlled Trials databases were searched. Only articles in English published up until 31 January 2024 were selected. Also, reference lists of the selected original papers and recent review articles were manually searched for additional sources. Data on study characteristics were extracted from the selected articles. Data on outcomes of interest were synthesized, and vaccine efficacy endpoints (histological lesion regression, clinical response, and overall survival) were selected as the basis for grouping the studies. Findings: After screening 831 articles, nine RCTs with 800 participants were included, of which seven studies with 677 participants involved CIN2 and CIN3 and examined lesion regression to ≤CIN1 as the efficacy endpoint. Results of two of these studies were deemed to have a high risk of bias, and another one did not contain statistical analyses. Results of the other four studies were quantitively synthesized, and the pooling of *p*-values revealed a significant difference between the vaccine and placebo groups in terms of lesion regression (*p*-values of 0.135, 0.049, and 0.034 in RCTs, yielding a combined *p*-value of 0.010). The certainty of the evidence was rated as moderate. Patients with advanced cervical cancers were studied in two RCTs with 123 participants. Clinical response and overall survival were taken as endpoints, and the results were reported as not significant. The certainty of the evidence of these results was rated as very low, mainly due to the very small number of events. All studies reported good tolerance for the vaccines. Conclusions and Relevance: The results indicate the potential for therapeutic vaccines in the regression of CIN2 and CIN3 lesions. Moreover, a potential gap in evidence is identified regarding the very low number of RCTs in patients with advanced cervical cancer.

## 1. Introduction

### 1.1. HPV-Associated Cervical Malignancies

Cervical cancer is the fourth most common cancer in females, with an estimated age-standardized incidence of 13.1 per 100,000 women in the world [1]. Infection with high-risk human papillomavirus (HPV) has been identified as the cause of this disease. Around 70% of these infections are caused by types 16 and 18 of HPV [2]. Cervical intraepithelial neoplasia (CIN) is the precursor to this disease. Most CIN lesions regress spontaneously, but some progress to cancer [3]. Moreover, these precancerous lesions may recur after regression or ablation.

### 1.2. Therapeutic Vaccines for HPV-Associated Cervical Malignancies

From the outset, the construction of prophylactic and therapeutic HPV vaccines faced similar challenges. HPV viruses could not be easily grown in the laboratory [4]. Therefore, unlike vaccines based on the attenuated or inactivated pathogenic organism, prophylactic and therapeutic HPV vaccines are made with recombinant proteins of specific components of the virus instead of the whole virus.

HPVs are non-enveloped, double-stranded DNA viruses with a high tropism for mucosal tissues. The genome comprises three regions: (i) a noncoding regulatory region; (ii) a region encoding for early-expressed proteins (E1, E2, E4–E7); and (iii) a late region related to genes that encode for the viral capsid proteins L1 and L2. More than 200 HPV genotypes have been identified. Oncogenic genotypes include HPV 16, 18, 31, 33, 35, 39, 45, 51, 52, 56, 58, 59, and 66.

Most HPV infections are transient and are eventually cleared by the body’s immune system, but a subset of high-risk (hr) HPV infections progress to clinical disease in the form of cervical intraepithelial neoplasia (CIN) and cervical cancer [5]. The key event in the process of oncogenesis is the increased expression of the viral E6 and E7 proteins in infected cells. E7 dissociates pRb from a transcription repression complex, thereby allowing the cell to shift from the G phase to the S1 phase of the cell cycle. E7 interacts with several other cellular proteins, including the STAT1 transcription factor, leading to significant transcriptional changes in the cell. This unchecked cell proliferation should trigger the protective mechanism of apoptosis, but as the E6 protein is also able to bind p53, a crucial tumor suppressor, the latter cannot play its role of accelerating programmed cell death. Moreover, E7 may cause errors in centromere duplication, resulting in genomic instability, and E6 activates telomerase, which is involved in maintaining telomere length and cell immortalization [6,7].

Several HPV proteins, namely L1, L2, E6, and E7, have been the focus of attention in vaccine development.

HPV viruses use L1 and L2 proteins to bind to the exposed basement membrane at the sites of microabrasion on the cervical surface and then to the epithelial cells [6,8]. Prophylactic HPV vaccines such as Gardasil^®^ (Merck, Kenilworth, NJ, USA) and Cervarix^®^ (GlaxoSmithKline, Brentford, UK) are virus-like particles (VLPs) that produce antibody responses to the L1 protein [9], thus blocking virus entry into the epithelial cells and preventing infection. However, these vaccines do not target established infections.

Having an important role in the process of oncogenesis and being expressed at high levels in malignant cervical cells, the E6 and E7 proteins are targets of therapeutic HPV vaccines [10]. Following their successful work on prophylactic HPV vaccine development, Frazer et al. conducted a clinical trial on a therapeutic HPV vaccine in patients with CIN in 2004. The vaccine was a recombinant E6E7 fusion protein [11]. In 2007, Kaufman et al. studied a VLP therapeutic vaccine in a clinical trial on subjects with CIN2 and CIN3 [12]. Subsequent clinical trials involved a range of approaches, including viral-vector, bacterial-vector, cell-based, and DNA-based vaccines.

### 1.3. Importance and Objectives of the Review

Given the burden of cervical cancer and the insufficiency of the established treatment modalities to cure it, therapeutic vaccines can be considered potential complements to the standard treatment. Recent years have seen a distinct increase in the number of RCTs conducted on these vaccines [13,14,15]. Thus, it is fitting to conduct a systematic review of the subject to examine the prospects of this treatment option in fighting cervical cancer and to identify possible gaps in the evidence. The present systematic review was conducted to assess the efficacy of therapeutic vaccines for HPV-associated cervical malignancies in women with cervical cancer or high-grade cervical dysplasia by synthesizing evidence from randomized clinical trials.

## 2. Methods

This systematic review was conducted according to the Preferred Reporting Items for Systematic Reviews and Meta-analyses (PRISMA) Statement guidelines [13].

### 2.1. Eligibility Criteria

Study inclusion criteria were determined according to PICOS (Participants, Intervention, Comparator, Outcome, and Study design) principles. Randomized controlled trials were included. Dose-escalation studies were included if they followed a randomized design. Studies on subjects with cervical cancer or advanced dysplasia of the cervix, variably referred to as cervical intraepithelial neoplasia (CIN) of CIN2 or CIN3 grades or high-grade cervical dysplasia (HSIL), were included. Studies involving several kinds of cancer were included. Vaccination with a therapeutic vaccine for HPV-associated malignancies was the experimental intervention eligible for inclusion in the review. Studies involving vaccines combined with other intervention modalities were also included. Eligible comparators included all forms of active and inactive control intervention. The outcome of interest was the efficacy endpoint of histological lesion regression, but other reported endpoints, including the clinical outcome, overall survival, immunological response, and clearance of HPV DNA, as well as adverse events and tolerability, were also addressed. Exclusion criteria encompassed HPV-seropositive subjects without pathological evidence of cervical neoplasia or cancer. Excluded studies are those on prophylactic vaccines against HPV for women and those initiated on patients in whom the lesion had been ablated through surgery or other interventions. The eligibility criteria were set by consensus among the authors.

### 2.2. Search Protocol

Search strategies were designed for specific databases according to the eligibility criteria described above. The PubMed/Medline, Embase, and Cochrane Central Register of Controlled Trials databases were searched for published and in-publication records. Only articles in English published up until 31 January 2024 were selected. In addition, we manually searched the references of review articles published from 2020 to 2024 that related to therapeutic vaccines for HPV-associated malignancies and immunotherapy for cervical cancer for other relevant papers.

### 2.3. Study Selection

After removing duplicates, titles, and abstracts, the retrieved records were screened, and records failing to meet the eligibility criteria were excluded. The full texts of the remaining records, including journal articles and meeting reports, were studied. Entries failing to meet the eligibility criteria were excluded, and the reasons for their exclusion were recorded. Lastly, papers and reports were mapped to their corresponding studies, and for each study, entries containing the most up-to-date and comprehensive information were identified; priority was given to peer-reviewed journal articles. The study selection process was performed in parallel and independently by two reviewers, and disagreements among the reviewers at any stage of this process were resolved through discussion.

### 2.4. Data Collection

General information on the studies and data regarding study characteristics (study design, participants and intervention groups, the type of vaccine evaluated and its comparators, efficacy outcomes, adverse events data and their analysis, and risk of bias) were extracted from the papers using a form modified from the Cochrane generic data collection form for RCTs.

### 2.5. Risk-of-Bias Assessment

The risk of bias in the included studies was assessed using the Cochrane risk-of-bias tool for randomized trials, version 2 (RoB 2) [14]. RCTs containing statistical analyses were included in the risk-of-bias assessment. The outcome of interest was the efficacy endpoint of lesion regression, defined as CIN ≤ 1 at the time of evaluation, and the summary measure was the difference in proportions. The risk-of-bias assessment process entailed assessing bias in five specific domains of randomization: deviations from the intended interventions, missing outcome data, measurement of the outcome, and selection of the reported results, as well as an assessment of the overall risk of bias in the study. The risk-of-bias judgments were given as one of three categories: low risk, some concern, and high risk of bias. The assessment of the results was visualized using the Robvis visualization tool [15].

### 2.6. Data Synthesis

Data on outcomes of interest were synthesized according to the Synthesis Without Meta-analysis (SWiM) guideline [16]. Other reported endpoints and adverse events were addressed in the narrative text.

Using the approach described by Anderson et al., a logical model was developed to guide the grouping of studies for synthesis [17]. Briefly, the model assumes that the intervention (vaccination) elicits the immune response, which results in the efficacy endpoint of lesion regression, with the ultimate efficacy endpoint being the overall survival. HPV DNA clearance is one of the intermediary changes linking the immune response to efficacy endpoints. Figure 1 depicts this model in diagram form. Based on this model, efficacy endpoints (lesion regression, clinical response, and overall survival) were selected as the basis for grouping the studies.

The significance level (*p*-value) for the outcome of interest (i.e., efficacy Endpoint) was used as the standardized metric in synthesizing the evidence. The statistical approach was to combine the *p*-values and the null hypothesis, which was the absence of vaccine efficacy in any of the trials considered. This metric was chosen because of the nature of the outcome of interest and the methods of its definition and measurement in the included studies. The pooling of the *p*-values was performed by Fisher’s method using the Stata statistical software (version 16). *p*-values < 0.05 were considered statistically significant.

### 2.7. Certainty of Evidence Assessment

Studies entailing statistical analysis on the outcome of interest were considered for the main synthesis. Studies containing direct evidence concerning the review question and judged to pose a low risk or some concern in the risk-of-bias assessment were prioritized. We explored heterogeneity in the reported effects visually using tables. The effect sizes in the grouped studies were compared, considering potential modifiers such as the vaccine type. The Grading of Recommendations, Assessment, Development and Evaluations (GRADE) approach was used in the assessment of the certainty of evidence [18]. All included studies were RCTs; therefore, we started the rating of the evidence with “high certainty” and downgraded it according to the level of concern in the risk of bias, inconsistency, imprecision, indirectness, and publication-bias domains.

## 3. Results

### 3.1. Search Results and Study Selection

Using database-specific search strategies, three databases, namely PubMed/Medline, Embase, and Cochrane Central Register of Controlled Trials, were searched, and only articles in English were selected from entries up to and including 31 January 2024. This search produced a total of 831 records. An additional 10 studies were identified through manual searching and added to this pool, which, after removing entries appearing in more than one database, was reduced to 703 records. We screened titles and abstracts of these 703 records and selected 68 for further assessment. In the full-text evaluation, 19 of these 68 records were deemed ineligible and excluded. The reasons for exclusion were as follows: most study participants harbored neoplasia other than cervical cancer (1 record), the study design was not an RCT (11 records), and the study was performed on subjects with low-grade dysplasia (CIN 1) or after lesion ablation (1 and 4 records, respectively). In two of the excluded records, the full text presented only the study protocol and did not contain outcome data. The remaining 49 records, including peer-reviewed journal articles and conference papers, were mapped to their respective studies. Thus, nine studies were identified for inclusion in the systematic review. Figure 2 details the study selection process in a PRISMA flow diagram.

### 3.2. Risk-of-Bias Assessment

Of the nine selected studies, seven included a statistical analysis on the efficacy endpoint and were assessed for the risk of bias [6,8]. Among these seven, two pioneering studies were judged to be at high risk of bias: one, because of missing outcome data, and the other, because of possible issues regarding the measurement of the outcome [11,12]. The other studies were of low risk in most of the assessed domains, and overall, they were judged to pose only some concern in terms of the risk of bias. Summary-level information on the risk of bias in the studies is presented in Figure 3, and the overall risk of bias across studies for each assessment domain is shown in Figure 4. Following RoB 2 guidelines, for each study, the authors’ judgment on the risk of bias in specific domains and support for the judgment were presented in conjunction with the characteristics of the study in Table 1 (characteristics of included studies).

### 3.3. Description of Included Studies

A total of nine RCTs, comprising 800 subjects, were included. In seven of these studies, the participants were subjects with advanced dysplasia of the cervix (CIN2, CIN3, or HSIL) [11,12,13,14,22,23,24], and in two studies, patients with refractory or recurrent cervical cancer were studied [19,22]. A range of therapeutic vaccine types, including long-peptide-based, [11,24], virus-like particle [12], viral-vector [21], bacterial-vector [22], cell-based [24], and DNA-based [20,23] vaccines, were investigated in these studies; however, all of them, except for one, involved the E6 and/or E7 component of HPV [19]. Trials focusing on advanced dysplasia of the cervix reported results on the efficacy endpoint of lesion regression [11,12,13,14,22,23,24,25], while trials studying patients with refractory/recurrent cancer of the cervix reported results on the efficacy outcomes of clinical response or overall survival [22,24]. Outcomes concerning the HPV DNA clearance and immune response, as well as data on adverse events, were also reported. Table 1 details the characteristics of the included RCTs. For each study, domain-specific and overall judgments on the risk of bias are also included.

### 3.4. Effects of Interventions

The included RCTs investigated several outcomes, including the efficacy of the studied vaccines in terms of their effects on disease-related endpoints (lesion regression, clinical response, and overall survival), HPV DNA clearance, and the immune response, as well as adverse events following vaccination. In this review, disease-related endpoints were taken as outcomes of interest, and studies were grouped accordingly. Table 2 presents the summary of these findings.

### 3.5. Lesion Regression

Of the nine RCTs, seven involved subjects with CIN2 or CIN3 and investigated lesion regression, defined as regression to the grade of CIN1 or complete resolution of the lesion, as the vaccine efficacy endpoint [11,12,13,14,22,23,24]. One of these studies did not contain any statistical analysis [19], and two were deemed at high risk of bias [11,12]; thus, four studies, comprising 598 subjects, were included in the quantitative evaluation [20,21,23].

Choi et al. investigated 72 patients with CIN3 and did not find a significant difference between the two arms of the interventions at the pre-specified time point. At week 20 after the intervention, 61% of 33 patients in the group receiving 1 mg of the vaccine showed histological lesion regression, while these numbers were 42% of 31 patients in the group who had received 4 mg of the vaccine (*p* = 0.135) [20].

Harper et al. studied 192 patients with CIN2 or CIN3 and found complete resolution in 24.0%, a partial response in 11.6%, and no response in 64.3% of patients in the vaccine group versus 9.5%, 11.1%, and 79.4%, respectively, in the placebo group (*p* = 0.049; calculated from data provided in the paper) [21].

Trimble et al. studied 169 patients with CIN2 or CIN3 and found lesion regression in 53 (49.5%) vaccine recipients and 11 (30.6%) participants in the placebo group (*p* = 0.034) [23]. The pooling of *p*-values revealed a combined *p*-value of 0.010. Based on the GRADE guidelines, the certainty of the evidence was evaluated as moderate.

Kawana et al. studied 165 patients with CIN2-3 and found complete regression in 12 (31.7%) out of 41 high-dose vaccine recipients and 5 (12.5%) out of 40 placebo recipients at week 24 (rate difference = 19.2, 95% CI = 0.5 to 37.8).

Two pioneering studies involving participants with CIN2 or CIN3 were deemed at high risk of bias because of missing outcome data and possible issues regarding the outcome measurement.

The RCT by Frazer et al. studied 31 subjects with CIN1-3 and did not show lesion resolution [11]. Likewise, Kaufmann et al. studied 39 patients with CIN2 or CIN3 and did not find a statistically significant difference between treatment groups in terms of the number of patients showing a more than, or equal to, 50% reduction in lesion size [12].

The study by De Vos van Steenwijk et al. comprised only a small number of subjects (nine patients) and did not include any statistical analysis. The researchers did not find lesion regression after vaccination [25].

### 3.6. Other Vaccine Efficacy Endpoints

Two studies focused on patients with advanced cancer; one of them investigated the clinical response, and the other study took overall survival as the vaccine efficacy endpoint.

Ramanathan et al. studied 14 patients with cervical cancer and reported a complete clinical response in 1 patient. However, the patient revealed that she had received chemotherapy after participating in the clinical trial [24].

Basu et al. studied 109 patients with recurrent/refractory cervical cancer. The researchers did not find a significant difference between treatment groups in terms of the median overall survival (OS); it was 8.28 months in the ADXS11-001 vaccine monotherapy group (95% CI = 5.85–10.5 months) and 8.78 months in the group receiving the ADXS11-001 vaccine combined with cisplatin (95% CI = 7.4–13.3 months) [22].

With regard to other outcomes and adverse effects, of the nine RCTs, six reported results on HPV DNA clearance and seven on the immune response. Most patients with histologic regression showed HPV clearance. Viral DNA clearance of CIN 2/3 was significantly greater in the vaccine-treated groups than in the placebo groups. Immune responses by specific T-HPV cells were more likely in vaccinated women. For Kawana et al., the number of HPV16–E7–specific-interferon-γ-producing cells increased with the level of the histological response.

Both outcomes were variably present, and the certainty of the evidence was rated as moderate.

Outcomes of clinical response and overall survival were reported in one study each (with 14 and 109 participants, respectively) [19,22]. Results for both outcomes were reported as not significant, and the certainty of the evidence was rated as very low, mainly due to the very small sample size and number of events. All studies examined adverse events and reported the vaccines as well tolerated with minor adverse events. The certainty of the evidence was rated as moderate.

## 4. Discussion

### 4.1. Summary of Main Results

A total of nine RCTs with 800 participants were included. The outcome of interest was the vaccine efficacy endpoint. Among the studies, seven focused on subjects with CIN2 and CIN3 and took lesion regression as the vaccine efficacy endpoint, while in two studies, participants were patients with advanced cervical cancer, and clinical response or overall survival was studied as the vaccine efficacy endpoint. The studies were grouped accordingly for synthesis. Table 2 summarizes the findings for the main comparison.

### 4.2. Lesion Regression

A total of seven studies, comprising 677 participants with CIN2 and CIN3, investigated lesion regression. In four RCTs with 598 participants, data on the outcome could be quantitatively synthesized. The comparison between experimental and comparator groups in terms of lesion regression in these studies showed *p*-values that were either not significant (*p* = 0.135 in the study by Choi et al.), marginally significant (*p* = 0.049 by Harper et al.), or that had a significance level near the cut-off point (*p* = 0.034 by Trimble et al.) [20,21,23]. Complete response was more pronounced in high-dose vaccine recipients than in the placebo group (rate difference = 19.2, 95% confidence interval [CI] = 0.5 to 37.8 in the study of Kawana et al.) [19]. Pooling *p*-values of the studies revealed a combined *p*-value of 0.010, indicating a clear effect of therapeutic vaccines on lesion regression in patients with CIN2 and CIN3.

### 4.3. Other Vaccine Efficacy Endpoints

Of nine studies, two RCTs involved 123 patients with advanced cervical cancer. One of these studies, with 14 participants, took the clinical response as the vaccine efficacy endpoint, and the other, with 109 participants, examined overall survival. None of these studies found a significant effect of the intervention on vaccine efficacy endpoints.

### 4.4. Comparison of Studies Grouped According to Efficacy Endpoints

Notable differences can be observed between the two groups of studies (lesion regression versus other vaccine efficacy endpoints) in terms of the number of studies (six versus two), the number of participants (512 versus 123), and the reported vaccine effect on the efficacy endpoints, i.e., marked effect on lesion regression in CIN2 and CIN3 versus no significant effect on clinical response or overall survival in patients with advanced cervical cancer.

### 4.5. Adverse Events

All studies reported the intervention as well tolerated. No major adverse event was reported.

### 4.6. Overall Completeness and Applicability of Evidence

RCTs on HPV therapeutic vaccines are still being actively developed and pursued. In the past three years alone, we witnessed one multicenter RCT per year [13,14,15]. Almost all RCTs conducted thus far have been in Phase I or Phase II. Thus, the available evidence on the subject is not complete yet. However, one potential gap in the evidence is already discernible. This gap was revealed as a result of grouping the studies based on the vaccine efficacy endpoint. The gap arises from the difference between the number of participants and clinical trials on cervical intraepithelial neoplasia and advanced cervical cancer, which, in the long-term, can lead to a paucity of high-quality evidence on the effect of HPV in recurrent/refractory cervical cancer, where it is sorely needed.

### 4.7. Certainty of the Evidence

In summary, the evidence included is based on nine RCTs with 635 participants.

### 4.8. Risk of Bias in and across Studies

The risk of bias in and across the studies was assessed according to the RoB 2 guidelines. Two pioneering studies were deemed at high risk of bias due to missing outcome data and possible issues regarding the measurement of the outcome [11,12]. In the remaining studies, the overall level of bias was judged to be at the level of “some concerns”. The concerns mostly involved the domains of the randomization process and deviation from the intended treatment. Strict adherence to specific methodological measures can obviate these risks and add to the rigor of the RCTs. Figure 3 and Figure 4 show summary results for the risk-of-bias assessment in and across the studies, respectively, and detailed RoB information is provided in Table 1.

### 4.9. Quality of Evidence

The quality of evidence was evaluated according to the GRADE guidelines. The rating began at the level of high quality, as all included studies were RCTs, and was downgraded after assessment. The quality of evidence for the outcomes of lesion regression (in CIN2 and CIN3) and immune response, as well as of adverse events, was rated as moderate, while the quality of evidence for the outcomes of clinical response and overall survival was determined to be very low. To summarize the rationale for these ratings, most studies were not at high risk of bias, and the possible risk was not likely to seriously impact the results. The level of heterogeneity in the studies was not high; even the vaccines, despite being of different types, were almost all based on the E6 and E7 components of HPV. Moreover, the constituents and comparisons of the RCTs were direct. The single factor that contributed to the downgrading was the imprecision arising from the relatively small sample size and number of events, resulting in a rating of moderate for most outcomes. In the case of the clinical response and overall survival outcomes, the quality rating was downgraded to very low because of the very small number of events. This rating means that further research is very likely to alter the results related to these outcomes. As for publication bias, PRISMA guidelines were followed to ensure that all relevant RCTs were included.

### 4.10. Limitations

#### 4.10.1. Potential Biases in the Review Process

The search strategies were devised so as to ensure that all eligible studies were included and to avoid relevant bias. They included only articles in English published up until 31 January 2024. Our study is not a meta-analysis; therefore, our results should be moderated by this fact [26].

The vaccines were experimental and elaborated by each team, but all used HPV 16-18 vaccines directed against oncogenic HPV. Nevertheless, the immune-effector mechanisms of these vaccines could be highly variable. This explains some differences concerning efficiency. However, the main common measured outcome for all RCTs was the histological regression of the lesions.

Another limitation of our study is the small number of included studies. Indeed, the present systematic review aimed to cover RCTs. This choice bolstered the rigor of the review but, at the same time, limited its scope to some extent.

#### 4.10.2. Agreements and Disagreements with Other Studies or Reviews

We did not find similar systematic reviews on therapeutic vaccines for HPV-associated cervical malignancies; however, recent reviews by Smalley et al. [27] on therapeutic vaccines in HPV-associated malignancies in general and Fakhr et al. on the immunotherapy of cancers caused by HPV are in agreement with our conclusions [28].

## 5. Conclusions

This systematic review found evidence of moderate certainty that therapeutic vaccines are effective in the regression of lesions of CIN2 and CIN3 grades. These results should be moderated by the fact that our study was not a meta-analysis. Thus, we need to explore the possible applicability of these vaccines in the management of pre-cancerous cervical lesions.

### 5.1. Implications for Practice

The study of therapeutic vaccines for HPV-associated cervical malignancies is a burgeoning field, and RCTs are likely to yield more robust evidence on the subject in the coming years. However, given the relative scarcity of RCTs on the use of therapeutic vaccines in patients with recurrent/refractory cervical cancer, clinicians and vaccine researchers can work together to devise protocols to integrate RCTs with the standard management of these patients in ethical and scientifically sound ways.

### 5.2. Implications for Research

The present systematic review identified a potential gap in the evidence, namely the low number of RCTs involving patients with advanced cervical cancer compared with RCTs involving patients with cervical intraepithelial neoplasms. This finding implies the need for efforts to address this imbalance. Furthermore, RCTs conducted thus far on patients with advanced cervical cancer have not reported significant results in terms of vaccine efficacy. Rating of the available evidence in the present systematic review indicated that further research is likely to change these results. However, researchers should keep an open mind on the subject. It is also possible that future RCTs will confirm the available evidence. Eventually, researchers might need to question the emphasis on the E6 and E7 components of HPV as the basis of therapeutic vaccines for advanced cervical cancer and re-examine this disease entity to search for other potential vaccine candidates. A new class of therapeutic vaccines based on messenger RNA (mRNA) will surely be evaluated in the upcoming years. Those vaccines can comprise multiple targets combined with mRNAs coding for immunomodulatory proteins. Recent studies by van der Jeught et al. [29], Grunwitz et al. [30], and Bever et al. [31] have shown very clearly the potential of mRNA as a therapeutic vaccine for HPV-associated cervical malignancies.

## Figures and Tables

**Figure 1 vaccines-12-00428-f001:**
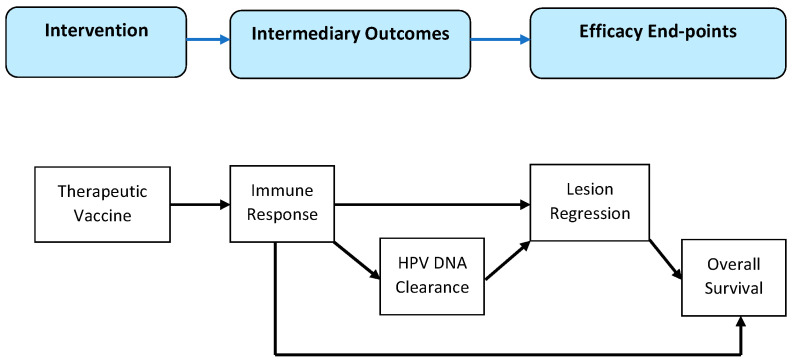
Logical model for grouping studies.

**Figure 2 vaccines-12-00428-f002:**
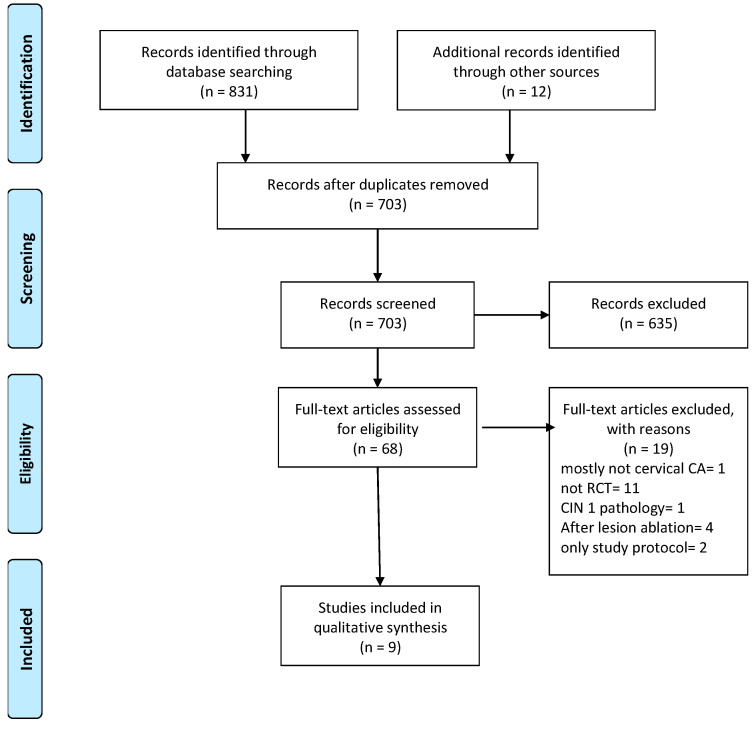
PRISMA flow diagram of the systematic review.

**Figure 3 vaccines-12-00428-f003:**
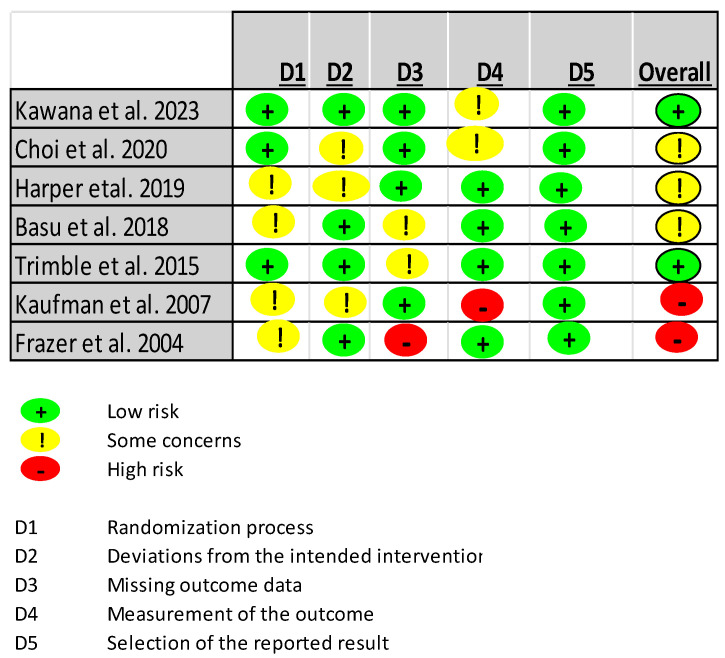
Summary information on the risk of bias in selected RCTs [11,12,19,20,21,22,23].

**Figure 4 vaccines-12-00428-f004:**
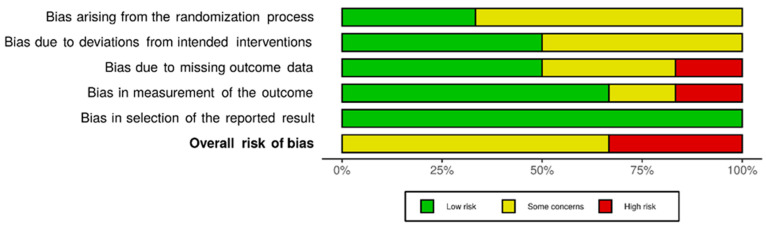
Overall risk of bias across selected RCTs for each assessment domain.

**Table 1 vaccines-12-00428-t001:** Characteristics of included studies.

Authors	Study Design	Participants	Interventions	Outcomes	Overall Bias
Kawana et al., 2023 [19]	Phase I and II, double-blind RCT	165 patients with HPV16-positive CIN2-3	IGMKK16E7 (*lacticaseibacillus paracasei* expressing cell surface, full-length HPV16 E7 was orally administrated to 4 groups: Placebo or low, intermediate, or high doses	Lesion regression, immune response, adverse events	Some concerns
Choi et al., 2020 [20]	Phase II, open-label RCT	72 patients with CIN2-3	GX-188E vaccine 4 mg, DNA vaccine (encoding HPV16 and HPV18 E6 and E7)	Lesion regression, HPV clearance, immune response, adverse events	Some concerns
Harper et al., 2019 [21]	Phase II, double-blind RCT	192 patients with CIN2 or CIN3	modified vaccinia Ankara (MVA) viral vector encoding IL-2 and HPV16 E6 and E7	Lesion regression, HPV clearance, adverse events	Some concerns
Basu et al., 2018 [22]	Phase II, RCT	109 patients with recurrent/refractory cervical cancer	ADXS11-001 monotherapy, Listeria monocytogenes containing the fusion protein Lm-LLO-E7	Overall survival, immune response, adverse events	Some concerns
Trimble et al., 2015 [23]	Phase II, double-blind RCT	169 patients with CIN2 or CIN3	VGX-3100 vaccine, DNA-based (mix of 2 plasmids encoding E6 and E7 genes from HPV16 and HPV18)	Lesion regression, HPV clearance, immune response, adverse events	Some concerns
Ramanathan et al., 2014 [24]	Phase I, RCT	14 patients with cervical cancer	cell-based (autologous tumor-lysate-primed mature dendritic cells)	Clinical response, adverse events	Not conducted (no statistical analysis)
De Vos van Steenwijk et al., 2012 [25]	Phase II, blinded RCT	9 patients with HSIL ^a^	HPV16 E6/E7 peptide	Lesion regression, immune response, adverse events	Not conducted (no statistical analysis)
Kaufmann et al., 2007 [12]	Double-blind RCT	39 patients with CIN2 or CIN3	CVLP vaccine 250 and 75 microg, virus-like particles (CVLP)- HPV16 L1E7 chimeric virus-like particles	Lesion regression, HPV DNA clearance, immune response, adverse events	High
Frazer et al., 2004 [11]	Phase I, double-blind RCT	31 subjects with CIN1, CIN2, or CIN3	HPV 16 E6E7 vaccine, 3 groups: 20 µg X3 60 µg X3 200 µg X1	Lesion regression, HPV DNA clearance, immune response, adverse events	High

^a^: HSIL—high squamous intraepithelial lesion.

**Table 2 vaccines-12-00428-t002:** Summary of findings for the main comparison: therapeutic HPV-vaccine effect on lesion regression in subjects with CIN2 and CIN3.

Outcome	Result	Number of Participants (Studies)	Certainty of the Evidence (GRADE)	Comments
Lesion regression to ≤CIN1	Combined *p*-value = 0.007	594 (4)	⊕⊕⊕⊝ Moderate	Kawana et al. [19], 2023, IC: 0.5–37.8 Choi et al. [20] 2020 (*p* = 0.135) Frazer et al. [11] 2004 (*p* = 0.12) Harper et al. [21] 2019 (*p* = 0.049) Trimble et al. [23] 2015 (*p* = 0.034)
HPV DNA clearance	Variably present	512 (6)	⊕⊕⊕⊝ Moderate	
Immune response	Variably present	687 (8)	⊕⊕⊕⊝ Moderate	
Clinical response	Not significant	14 (1)	⊕⊝⊝⊝ Very low	
Overall survival	Not significant	109 (1)	⊕⊝⊝⊝ Very Low	
Adverse events	Well tolerated	796 (9)	⊕⊕⊕⊝ Moderate	
**GRADE: Working group grades of evidence**				
High quality: Further research is very unlikely to change our confidence in the estimation of the effect				
Moderate quality: Further research is very likely to have an important impact on our confidence in the estimation of the effect and may change the estimated effect				
Very low quality: Very little confidence in the effect estimated				

## Data Availability

The data can be shared up on request.

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
