# Peer review of "Therapeutic Vaccines for HPV-Associated Cervical Malignancies: A Systematic Review"

_vaccines, 2024, doi:10.3390/vaccines12040428_

Round 1

Reviewer 1 Report

Comments and Suggestions for Authors

Estimated Authors,

I've been invited to review your Systematic Review entitled "Therapeutic Vaccines for HPV-Associated Cervical Malignancies". In your study, a total of 6 primary studies were included: in accord with your results, therapeutic vaccines can be acknowledged as effective in the regression of lesions of CIN2 and CIN3 grades. Unfortunately, as your summary is only a qualitative one, without any meta-analysis of included study, your results should be acknowledged with required caution (that should be advocated even in the discussion and conclusion section). Similarly, Authors should include a specifically designed "limits" subsection where they should include all of the main issues associated with the design of their study, including the limited number of primary studies eventually included and the lack of quantitative analysis.

Regarding the content of the paper, I've neither major concerns or doubts. On the other hand, please note some awkward design in both Table 1, that is quite difficult to be followed regarding its content, and the overall organization of the main text, with several subsection that should be numbered according to the instructions to the authors of Vaccines.

Author Response

Answers to comments Reviewer 1

Dear Reviewer

Thank you for your comments.

We introduced all suggested modifications

Sincerely

Unfortunately, as your summary is only a qualitative one, without any meta-analysis of included study,

your results should be acknowledged with required caution (that should be advocated even in the discussion and conclusion section). OK added in the discussion in « limits » and conclusion: Answer :Our study was not a metanalysis and therefore our results should be moderated by this fact.  Another limitation of our study is the small number of included studies.

Similarly, Authors should include a specifically designed "limits" subsection where they should include all of the main issues associated with the design of their study, including the limited number of primary studies eventually included and the lack of quantitative analysis answer :a subsections limits if the study was added in the end of the discussion :

. On the other hand, please note some awkward design in both Table 1, that is quite difficult to be followed regarding its content, and the overall organization of the main text, Answer: Ok the table 1  was simplified please see in the results table 1

with several subsection that should be numbered according to the instructions to the authors of Vaccines.Answer: ok all sections and subsections are numbered

Reviewer 2 Report

Comments and Suggestions for Authors

In the present systematic review, the Authors aimed to investigate the effects of therapeutic vaccines in the treatment of patients with cervical cancer and cervical intraepithelial neoplasia (CIN) grades 2 and 3, based on the published randomized controlled trials (RCTs). Even though the data obtained in the study could be of value for the future studies, I have some major comments for the Authors’ consideration.

Major comments

1) In the Abstract and Discussion sections of the manuscript it is stated that the literature search was performed without date and language restrictions. However, in the Methods and Results sections of the manuscript it is stated that only articles in English, published until January 31 (Methods) or February 1 (Results), were selected. Please correct the inconsistencies. 

2) In addition, as only articles up to the beginning of 2021 were included in the study and as this field is rapidly evolving, the literature search should be repeated to include possible novel articles, published at least until January 2024.

Minor comments

1) In Table 2 it is not clear which studies were included to investigate individual outcomes and the meaning of “very low GRADE” is not explained.

2) Please make sure that the used nomenclature and symbols are concise throughout the manuscript (e.g. p-value, P) and that repetition is avoided (e.g. lines 133 and 135).

3) Please make sure that the manuscript is thoroughly proofread by the English native speaker.

Comments on the Quality of English Language

Please make sure that the manuscript is thoroughly proofread by the English native speaker.

Author Response

Comments of Reviewer 2

Dear reviewer,

 Thank you for your comments.

We revised our article according to your comments.

Sincerely

Major comments

1) In the Abstract and Discussion sections of the manuscript it is stated that the literature search was performed without date and language restrictions. However, in the Methods and Results sections of the manuscript it is stated that only articles in English, published until January 31 (Methods) or February 1 (Results), were selected. Please correct the inconsistencies. Answer Yes we delete the mistake and replaced it by:: Only articles in English until 31 January 20244 were selected in the Abstract, Method and Dsicussion

2) In addition, as only articles up to the beginning of 2021 were included in the study and as this field is rapidly evolving, the literature search should be repeated to include possible novel articles, published at least until January 2024. Answer ok: we completed the literature search until 31 January 2024, we add new RCT and included new RCT who fill inclusion and exclusion criteria

Minor comments

1) In Table 2 it is not clear which studies were included to investigate individual outcomes and the meaning of “very low GRADE” is not explained. Answer Added in the legend table 2 : TVery low quality; Very little confidence in the effect estimated

2) Please make sure that the used nomenclature and symbols are concise throughout the manuscript (e.g. p-value, P) and that repetition is avoided (e.g. lines 133 and 135). Answer ok corrected

3) Please make sure that the manuscript is thoroughly proofread by the English native speaker Answer the manuscript was revised by an English native speaker

Round 2

Reviewer 2 Report

Comments and Suggestions for Authors

I have no further comments.